# Insight into Oncogenic Viral Pathways as Drivers of Viral Cancers: Implication for Effective Therapy

Ahmed M. E. Elkhalifa [1,2,*], Showkat Ul Nabi [3], Ovais Shabir Shah [4], Showkeen Muzamil Bashir [5], Umar Muzaffer [6], Sofi Imtiyaz Ali [5], Imtiyaz Ahmad Wani [7], Nasser A. N. Alzerwi [8], Abozer Y. Elderdery [9], Awadh Alanazi [9], Fawaz O. Alenazy [9] and Abdulaziz Hamdan A. Alharbi [10]

1. Department of Public Health, College of Health Sciences, Saudi Electronic University, Riyadh 11673, Saudi Arabia
2. Department of Haematology, Faculty of Medical Laboratory Sciences, University of El Imam El Mahdi, Kosti 1158, Sudan
3. Large Animal Diagnostic Laboratory, Department of Clinical Veterinary Medicine, Ethics & Jurisprudence, Faculty of Veterinary Sciences and Animal Husbandry, Sher-e-Kashmir University of Agricultural Sciences and Technology, Srinagar 190006, Jammu and Kashmir, India
4. Department of Sheep Husbandry Kashmir, Government of Jammu and Kashmir, Srinagar 182301, Jammu and Kashmir, India
5. Biochemistry & Molecular Biology Lab, Division of Veterinary Biochemistry, Faculty of Veterinary Sciences and Animal Husbandry, Sher-e-Kashmir University of Agricultural Sciences and Technology, Srinagar 190006, Jammu and Kashmir, India
6. Department of Medicine, Government Medical College and Associated Hospital, Srinagar 190010, Jammu and Kashmir, India
7. Clinical Research Laboratory, SKIMS, Srinagar 190011, Jammu and Kashmir, India
8. Department of Surgery, College of Medicine, Majmaah University, Ministry of Education, Al Majmaah 11952, Saudi Arabia
9. Department of Clinical Laboratory Sciences, College of Applied Medical Sciences, Jouf University, Sakaka 72388, Saudi Arabia
10. Hail Regional Laboratory, Hail 55471, Saudi Arabia
* Correspondence: a.alkhalifa@seu.edu.sa

**Abstract:** As per a recent study conducted by the WHO, 15.4% of all cancers are caused by infectious agents of various categories, and more than 10% of them are attributed to viruses. The emergence of COVID-19 has once again diverted the scientific community's attention toward viral diseases. Some researchers have postulated that SARS-CoV-2 will add its name to the growing list of oncogenic viruses in the long run. However, owing to the complexities in carcinogenesis of viral origin, researchers across the world are struggling to identify the common thread that runs across different oncogenic viruses. Classical pathways of viral oncogenesis have identified oncogenic mediators in oncogenic viruses, but these mediators have been reported to act on diverse cellular and multiple omics pathways. In addition to viral mediators of carcinogenesis, researchers have identified various host factors responsible for viral carcinogenesis. Henceforth owing to viral and host complexities in viral carcinogenesis, a singular mechanistic pathway remains yet to be established; hence there is an urgent need to integrate concepts from system biology, cancer microenvironment, evolutionary perspective, and thermodynamics to understand the role of viruses as drivers of cancer. In the present manuscript, we provide a holistic view of the pathogenic pathways involved in viral oncogenesis with special emphasis on alteration in the tumor microenvironment, genomic alteration, biological entropy, evolutionary selection, and host determinants involved in the pathogenesis of viral tumor genesis. These concepts can provide important insight into viral cancers, which can have an important implication for developing novel, effective, and personalized therapeutic options for treating viral cancers.

**Keywords:** virus; oncogenes; pathogenesis; genome; transition; metabolic; bio-thermodynamics; cancer

## 1. Introduction

Currently, much of the collective research efforts across the world have been directed at understanding cancer pathogenesis and pathways involved in carcinogenesis [1]. Cancer Genome Atlas (TCGA) Pan-Cancer project has developed a database of numerous types of cancers. They have reported mutation in essential genes as the significant determinant for the development of cancer [2]. Recently, the World Health Organization (WHO) has postulated that among different types of cancer, 15.4% are caused by infectious agents, and 9.9% of cancers are attributed to various types of viruses collectively called oncogenic viruses [2,3]. Furthermore, studies have indicated that the incidence of viral cancers is highest compared to different cancers. Going back to 1991, the first Cancer-causing virus "Rous sarcoma virus (RSV)" was found to cause cancer in chickens, and this led to the addition of a new field in cancer biology called "viral oncology" indicating biotic causes of cancer and with this extensive search for another virus with cancer-causing potential was undertaken [4]. The International Agency for Research on Cancer (IARC) has identified eleven pathogens as potentcancer-causing agents, and among them, Helicobacter pylori occupy the top position [5]. Multiple research groups worldwide have extensively researched and developed novel approaches against viral cancers. For instance, recently, in 2020, a novel type D beta-retrovirus, Gunnison's prairie dog retrovirus (GPDRV), for its role in thymus cancer in prairie dogswere identified [6]. Canine papillomavirus (CPV) causes squamous cell carcinoma in canines. Genomic analysis has revealed that the E2 protein and chimeric proteins E8 and E2 of CPV act as inducers for oncogenes [7].

Similarly, after extensive research to evaluate promoters of cervical cancers in Mexican women, researchers have found a high viral load of human papillomavirus (HPV) in women's reproductive tract, indicating a significant role of chronic HPV infection in cervical cancer [8]. These viral families were found to cause activation of Janus kinase/signal transducer and transcription (JAK/STAT) signaling pathways and hence promote cervical carcinogenesis [9]. Furthermore, Epstein–Barr virus (EBV) offers a classic example of cancer-inducing viruses.

In preclinical studies, latent membrane protein 1 of EBV (EBV-LMP1) was found to cause upregulation of mitochondrial enzymes, which includes glutaminase-1 (GLS1), isoforms of kidney-type glutaminase (KGA) and glutaminase-C (GAC) that boosts the pathogenesis of classical Hodgkin's lymphoma (cHL) [10–12]. In addition to uncovering the role of EBV in the pathogenesis of cHL, these studies indicate that effective blocking of GLS1 and GAC can provide potential treatment against EBV-associated cancers [13]. Merkel cell polyomavirus (MCPyV) has been found to have a pathogenic role in Merkel cell carcinoma (MCC), a life-threatening skin cancer [14]. In addition, it has been found that these viruses play a role in cancer pathways. Hence, it provides an opportunity to develop a vaccine for the prevention/reduction of the cancer burden. Furthermore, there is evidence that supports the role of viral oncogenes in the pathogenesis of breast cancer and colorectal cancer (Figure 1) [15]. Figure 1 provides a classical example of cancer induction by viral oncoproteins in the world's most common types of cancers.

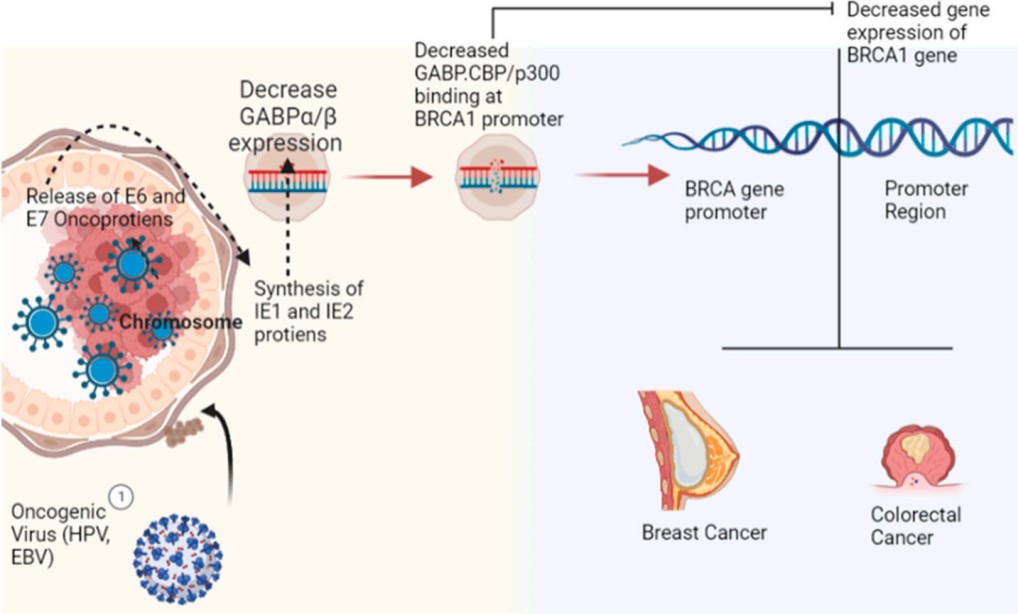

**Figure 1.** Role of viral oncogenic proteins in the progression of the most common type of cancers (breast cancer and colorectal cancer) with particular emphasis on the BRCA1 gene (Image created by biorender.com, accessed on 2 November 2022).

## 2. Basic Characteristics of Viral Cancers

Owing to the experimental support for the role of viruses in carcinogenesis, researchers have postulated some of the essential characteristics for identifying the role of the virus in cancer pathogenesis [16]; these include (i) Persistent co-existence of cancerous cells and viral particles in cancer biopsy (ii) Growth promoting activity of viral genes (iii) Expression of malignant phenotype promoted by the presence of viral constituents which cause modification of host genomic constituents (iv)Epidemiological evidence to support the role of the virus in cancer prevalence. The most frequent form of co-existence between an oncogenic virus and cancerous cells is the presence of viral particles as a persistent infection at the site of tumor genesis (HCV) [17]. However, the existence of viral particles as extrachromosomal episomes (MCPyV) and viral genetic Integration into the host genome (HTLV-1) are two additional forms reported by researchers [18]. Integration of the viral genome with the genetic components of host cells results in the mutation of the host cell genome that is passed from one cell generation to the following [19]. The co-existence of viral particles as chronic infection offers a unique mechanism for viral carcinogenesis, as these viruses do not directly harm host cells. Still, their presence in the cellular milieu results in low-grade chronic inflammation that catalyzes the development of cancer [20,21].

## 3. Types of Viral Cancers

Oncogenic viruses can be divided into direct and indirect oncogenic viruses. Indirect oncogenic viruses cause chronic and sustained alterations in cellular and subcellular pathways that lead to the clinical endpoint of cancer by activating alternative pathways, for example, inflammatory pathways. In contrast, direct oncogenic viruses use intracellular viral components to drive the cancer pathway [22]. Direct viral oncogenesis is characterized by the interaction between the immune system and viral infections; immune checkpoint suppression appears to be the only factor linking these tumors to developing viral cancers [22]. Most oncogenic viruses identified by IARC have been categorized under Group I carcinogens for humans (Table 1).

**Table 1.** Viruses are classified as group I carcinogenic viruses per IARC, a summary of carcinogenic determinants and associated cancers caused by these oncogenic viruses.

| Virus | Oncogenic Cancer-Associated | Oncogenic Viral Products | Mechanism Involved | References |
|---|---|---|---|---|
| EBV | Hodgkin's lymphoma, Gastric Cancer, Lymphoepithelioma | circular RNAs (circRNAs), EBV circular BamHI A rightward transcripts (circBARTs), EBV latent membrane protein 1 (EBV LMP1) | Various genes and signaling pathways influence the development of EBV-related neoplasms. Activate oncogenes such as Bcl-2 and MYC and signaling pathways, including NF-B, JNK, JAK/STAT, and PI3K/Akt, and deactivate tumor suppressors such as p53, p27kip1, p21WAF1/CIP1, p16INK4A, p73, PRDM1, DICE1, and p27kip1. Methylation modification, evasion of apoptosis, alterations in the transcription of DNA methyltransferases, suppression of anti-tumor genes (p16 and p53) | [11,23–27] |
| HHV-8 | Kaposi's sarcoma, Multicentre Castleman's Disease, Primary Effusion Lymphoma | cirRNAs, vIRF4 viral locus (circvIRF4), RNase-R resistant Polyadenylated Nuclear (circPAN), non-coding RNA, Latency Associated Nuclear Antigens | Alteration in Notch pathways and hypermethylation followedby slower hypomethylation, upregulation of Protocadherin Beta-5 (PCDHB5), | [23,28–30] |
| HTLV-1 | leukemia/ lymphoma | Tax proteins, HTLV-1 Tax protein, translocation of methylcytosine dioxygenase genes | proviral integrations on chromatin loops, disruption of transcriptional pathways, inactivation of p53, and hypermethylation at oncogenic promoter regions. | [1,29,31,32] |
| HBV | Pancreatic Carcinoma, Hepatocellular carcinoma, | short mRNAs, latent membrane protein 1 (EBV LMP1), Latency Associated Nuclear Antigens | chromosomal instability, upregulation of Small Protein of the HBV Surface Antigen, enhanced cell migration, Methylation mechanisms | [13,33–35] |
| HIV-1 | Kaposi's sarcoma, skin Carcinoma, Hepatocellular carcinoma, cancer of the conjunctiva | Intrinsically Disordered Proteins (IDPs) | Suppression of p16 expression, immune dysregulation, and immune evasion, there are five viral proteins: transactivator of transcription, accessory protein negative factor Nef, matrix protein p17, and envelope protein gp120. Reverse transcriptase RT and Tat. When secreted from HIV-1-infected cells, Gp120, Nef, p17, Tat, and RT produce oxidative stress and are carcinogenic | [33,36–39] |
| HPV | Cervical Cancer, oral Cancer, tonsillar carcinoma, penile Cancer | endogenous protein E6 and E7, retinoblastoma protein, HPV6, 11, 16, and 18, Dedicator of cytokinesis-8 (DOCK-8), Alpha-7 genotypes of HPV | Inactivation of p53, DNA methylation, methylation modifications, and Integration serves as a precursor to the passage from LSIL to HSIL, Oxidative stress resulting in DNA damage | [26,28,40–45] |
| HCV | Non-Hodgkin's lymphoma, Gallbladder carcinoma, Thyroid carcinoma | E1 and E2 membrane proteins, non-coding RNA, histone modification, transcriptional alteration | Immune evasion, signal transducer and activator of transcription 3 (STAT-3) activation, suppression of DNA methyltransferases activity, suppression of tumor suppression genes, Genomic hypomethylation of apoptosis genes. | [17,19,46] |
| MCPyV | T-cell leukemia/lymphoma and Merkel Cell Carcinoma | large T antigen (LT), Full-length LT, Truncated LT antigen mutation (tLT), small T antigen (ST) | cell immortalization, cell Transformation, immune evasion, Onco-suppressive proteins (Rb), downregulation of Toll-like Receptor-9 (TLR-9) | [17,27,47,48] |
| HPV E6 | Cervical Cancer Breast Cancer Colorectal Cancer | cellular proteins(p21 and pRb) deregulates expression of p53 and BCL2 E6 stimulates cell proliferation independently from E7 through its C-terminal PDZ-ligand domain | E6 degrades p53, targets c-myc oncogene (a marker protein for several cancer forms including cervical cancer), Inactivates p53, and releases repression of BCL2Mediates suprabasal cell proliferation and disrupts normal cell adhesion to contribute to the development of metastatic tumors | [37,49–51] |

| Virus | Oncogenic Cancer-Associated | Oncogenic Viral Products | Mechanism Involved | References |
|---|---|---|---|---|
| HPV E7 | Cervical Cancer Breast cancer Colorectal Cancer | Interacts with DREAM (dimerization partner, RB-like, E2F4, and MuyB) BRCA1 gene and dissociates the pRB-E2F complex by binding to pRB | Inhibits retinoblastoma protein (pRb)epigenetic derepression through KDM6B (H3K27-specific) Demethylase 6B triggers the expression of p16INK4A, targets c-myc oncogene, lowers p53, and increases BCL2 levelExpresses proteins necessary for DNA replication | [25,37,52–54] |

Researchers worldwide have contributed to comprehensive genome and whole-transcriptome analyses of cancerous samples that have collected information about human cells and other cancer-causing pathogens [55]. International Cancer Genome Consortium (ICGC) has developed a database that contains comprehensive whole-genome and whole-transcriptome data to provide an opportunity for in-depth research on cancer viruses. These studies have found a significantly distinct expression profile of cancerous samples compared to normal samples. According to ICGC, the most critical factor for cancer progression is the integration of the viral genome with the host genome [56]. For instance, HPV E7Inhibits retinoblastoma protein (pRb) epigenetic derepression through KDM6B (H3K27-specific Demethylase 6B to trigger the expression of p16INK4A, targets c-myc oncogene Lowersp53 and increases BCL2 level Expresses proteins necessary for DNA replication (Table 1). Similarly, MCPyV causes cell immortalization, cell Transformation, and immune evasion, which plays a pivotal role in Cervical Cancer, Breast Cancer and Colorectal Cancer (Table 1).

## 4. Possible Role of COVID-19 Infection in Cancer Progression

COVID-19 infection is considered a serious health problem and has caused heavy mortality in cancer patients and patients with other comorbidities [57–59]. It has been postulated that severe acute respiratory syndrome coronavirus 2 (SARS-CoV-2) induces autophagy and leads to the progression of cancer [60]. Furthermore, it has been found that SARS-CoV-2 infection causes drug resistance and the ineffectiveness of immune therapy in cancers. Interestingly, various coronaviruses induce autophagy, and the pathway is very complicated and has yet to be fully elucidated [60]. Coronaviruses employ these autophagy processes for viral multiplication. Recently, it has been reported that SARS-CoV-2 induces structural alterations in the endoplasmic reticulum and causes the synthesis of blisters filled with viral RNA. These scaffolds protect viruses from the host immune system and inhibit viral clearance [61]. In addition, it has been found that ORF8 protein encoded by SARS-CoV-2 causes activation of BECN1 that results in lysosome degradation of major histocompatibility complex I (MHC-I) molecules. Another study has found that the concentration of SQSTM1 increases up to 1.5-fold in SARS-CoV-2 infection, and SQSTM1 has been found to play a pivotal role in the formation of autophagosomes [62,63]. In addition, to induce autophagy, SARS-CoV-2 causes cellular alterations, including hypoxia, oxidative damage, cytokine storm, and increased levels of pro-inflammatory cytokines. Such conditions have been reported to cause various cancers [17,64].

In earlier studies, it has been reported thatSARS-CoV-2 induced hypoxia causes activation of HIF-1$\alpha$ which is an essential biomarker of acute myeloid leukemia (AML), especially in those patients who haveFlt3-ITD (FMS-like tyrosine kinase-3 receptor internal tandem duplications) mutations. In vitro studies conducted by [65,66] found that hypoxia, cytokine storm, and generation of reactive oxygen species result in activation of NF-$\kappa$B, which stimulates autophagy and subsequently causes loss of Caveolin-1 (CAV-1)from the cellular microenvironment that can lead to tumor recurrence. It has been reported that regulation of AMP-activated protein kinase (AMPK)plays an important role inthe overexpressionof proline oxidase (POX),responsible for the conversion of proline into pyrroline-5-carboxylate

(P5C), which in turn isinvolved in cell survival, apoptotic cell death, and autophagy in cancer cells [67–70]. Furthermore, POX accelerates autophagy through increased production of free radicals and enzymatic cleavage of mTOR (as an autophagy suppressor) [28]. Autophagy helps in the proliferation of cancerous cells in a nutrient-depleted environment; hence it seems that SARS-CoV-2 promotes carcinogenesis by initiating autophagy pathways [71].

Furthermore, many in-vitro studies have found that autophagy-related drugs can serve as potential therapeutic drugs against SARS-CoV-2 [72]. In addition, it can be postulated that SARS-CoV-2 induces autophagy that can result in cancer progression; hence targeting autophagy can be an effective therapeutic strategy against SARS-CoV-2-induced cancer. In SARS and MERS, comorbidities play a significant role, and many risk variables are linked to poor illness outcomes, particularly advanced age, and male sex. Cancer and co-infections are additional MERS risk factors for a bad prognosis. Although research in this area is still in its nascent stage, there is an urgent need to evaluate this hypothesis with further studies [22,31].

## 5. Multiple Mechanisms of Viral Carcinogenesis

In cancer biology, there are numerous mechanisms and pathways involved in carcinogenesis. Early studies in this area have identified a limited number of cancer pathways involved in viral oncogenesis. Still, with the advancement of research, more pathways and complexity have been found in viral cancer pathogenesis [73]. Although, there are well-recognized pathways identified for cancers caused by oncogenic viruses. For instance, EBV activates oncogenes such as Bcl-2 and MYC and deactivates tumor suppression genes [23,24]. Similarly, HHV-8 causes dysregulation of Notch pathways and hyper methylation of Protocadherin Beta-5 (PCDHB5), which has been identified to play a significant role in cancer of various types [18,54]. Furthermore, HTLV-1 has been found to integrate with the host genome and cause disruption of chromatic structure, which brings structural alteration in chromosomal structure and brings promoter regions in proximity to dormant oncogenes [72]. In continuation with these findings, HBV causes chromosomal instability and upregulation of small latent proteins [13]. In the subsequent section, we attempted to explore the biological processes involved in the pathogenesis of cancers caused by oncogenic viruses and their role in the activation of various cancer pathways by acting on multiple targeting sites of cancer pathogenesis.

The first animal model investigation on carcinogenic virus discovered the viral genome to encode the oncogenes v-src, v-myc, and v-ras that transform healthy developing cells into malignant cells. Subsequently, these genes were found to be involved in other cancers and were found to be dysregulated in cancer cells [47]. It has been reported that DNA tumor viruses translate novel oncoproteins such as SV40 T-antigen proteins, E1A and E1B proteins from adenovirus, and E6 and E7 from papillomavirus, which inhibit tumor suppressor proteins (p53 and Rb) to cause changes inthe microcellular environment for the promotion of cancerous growth [74]. Although these oncogenes create a favorable environment for carcinogenesis, the co-existence of some additional necessary risk factors is also required that can explain the rare occurrence of cancer in the majority of the population infected with the virus having a carcinogenic potential [75]. Identification of limited cellular targets of viral tumor genesis and their unsatisfactory explanation to understand the pathogenic pathways involved led the researchers to search for other alternative and additional pathways involved in viral oncogenesis [76]. For instance, it is well known that the viral oncoprotein E7 binds to and inhibits the competitive regulation of Rb cell cycle control, weakening the cellular protective framework against cancer. However, it has been studied that the same oncoprotein interacts with the Rb-associated DREAM complex, phosphatase PTPN1, histone-modifying enzyme HDACs, stem cellpromoting factors (APH1B and OCT04),and Cullin-2 to stabilize APOBEC3a [77]. In addition to the activation of these pathways, the oncoprotein E7 binds to the other oncoprotein E6, and their adduct triggers the activation of hTERT, resulting in accelerated and uncontrolled

cellular growth [58]. Therefore, it is possible to hypothesize that viral subtypes and the microenvironment of the host tissue determine the translation of these interactions in oncogenesisand the potential of a single oncoprotein to interact with multiple hotspots of cancer pathways [25].

Cancers caused by viruses may not appear immediately after infection; instead, they appear 15 to 40 years later [78]. The rare EBV-associated lympho proliferative illness, which can develop soon after infection [78], is an exception. Viral replication is either missing or significantly reduced in malignancies [33,78] because active replication would lyse the host cell and stop carcinogenesis. The virus is presented intracellularly as a naked nucleic acid in the form of a plasmid, episome, or genome integrated into the host cell [33]. RNA viral genomes must be reverse-transcribed into DNA before integration can occur, whereas DNA virus genomes can integrate straight into the host genome [79]. The two most crucial elements of this evolutionary analysis are the size of the coding capacity and the characteristics of the DNA-dependent DNA polymerase enzyme encoded by the large DNA viruses. Based on the following factors, this analysis prediction, and the evaluation of the DNA viruses' oncogenic potential will be made: (i). The DNA viruses' carcinogenic potential and genome size would be inversely correlated. The likelihood of large DNA viruses causing cancer is significantly higher than small DNA viruses. This argument is based on the observation that large DNA viruses can infect particular host cells latently for a lifetime as opposed to DNA viruses with a smaller genome [80]. (ii). Cancer cannot be developed by DNA viruses that exclusively infect their natural hosts with lytic infections. Lytic infections result in cell death, eliminating the possibility of the malignant transformation that is frequently brought on by persistent infections of a cell lineage [80]. This is conceivably the most significant prediction since it contributes to the solution of a complicated viral oncology riddle. Long-standing research has shown that viruses such as the JC virus and the BK virus induce cells to change malignantly when cultured in the laboratory [81]. Therefore, constant infection of a cell lineage is necessary for virus-induced carcinogenesis. However, these viruses have the potential to cause cancer if they unintentionally integrate into the host's genome or if oncogenes are continuously expressed in cells that are not receptive. This occurs in the case of Merkel Cell Carcinoma, which is brought on by skin infection with the polyomavirus [82,83]. Similarly, MCPyV of polyomavirus, being a small oncoprotein, binds explicitly to T cells and alters the transcriptional regulatory framework of the cell cycle, and causes activation of protein phosphatase 2Awhich results in the production of the F-box protein FBW7, which indirectly activates the NF-kBsignaling pathway and increases the production of free radicals that damage the cellular framework of proteins and nucleic acids [77]. These multitasking pathways result in the activation of oncogenes and hence the progression of cancer activated by viral oncoproteins [83].

Furthermore, DNA viruses such as EBV and KSHV cause the synthesis of a diverse set of viral oncoproteins, and these proteins precisely target the cellular host machinery [84]. A variety of genes influences the development of EBV-related neoplasms and signaling pathways, which includes activation of oncogenes such as Bcl-2 and MYC, as well as signaling pathways including NF-B, JNK, JAK/STAT, and PI3K/Akt, and deactivate tumor suppressors such as p53, p27kip1, p21WAF1/CIP1, p16INK4A, p73, PRDM1, DICE1, and p27kip1(I, II) [14]. For instance, EBV causes the synthesis of two membrane-binding proteins (LMP1 and LMP2), and these proteins mimic the functioning of the B-cell receptor and CD$^{4+}$ Co-receptor, henceforth causing immobilization and apoptosis of B lymphocytes [85]. Additionally, LMP1 binds to TRAFs and TRADDs, which causes activation of the NF-kB pathway, whereas LMP2 binds to src family kinases, leading to the synthesis of nuclear antigens (EBNA-LP) [70,86,87], which are supposed to be highly carcinogenic. Similar to EBV, KSHV promotes the production of the nuclear antigen LANA, which inhibits p53 and Rb activity and subsequently causes up regulation of vGPCR, which culminates in endothelial tumor genesis [88]. In addition to these coding proteins EBV and KSHV contain many non-coding RNAs which have a significant role in carcinogenesis. For instance, EBV

containing EBERs (a non-coding RNA) interacts with awide array of cellular pathways, such as transcription factors, ribosomal subunits, and TLR receptors that serves as paracrine signaling for the proliferation of cells [22]. Furthermore, recent studies have found the role of viral mi-RNA in disease progression to exemplify EBV-associated gastric carcinomas (EBVaGC). EBVaGCwas found to have high levels of miRNAs synthesized from the same locus (BARTs) from where carcinogenic non-coding RNA is synthesized, henceforth potentiating the carcinogenic effect in combination with other mediators of cancer [29]. The development of EBV-related neoplasms is influenced by a variety of genes and signaling pathways, which include activation of oncogenes such as Bcl-2 and MYC, as well as signaling pathways including NF-B, JNK, JAK/STAT, and PI3K/Akt, and deactivation of tumor suppressors such as p53, p27kip1, p21WAF1/CIP1, p16INK4A, p73, PRDM1, DICE1, and p27kip1(I, II). Interestingly, these viral determinants are expressed heterogeneously in different individuals, complicating viral carcinogenesis and challenging comprehending viral oncogenic pathways [65]. In the following section, we have attempted to discuss this expanding list of viral oncogenic pathways.

### 5.1. Inhibition of Apoptosis

Inhibition of apoptosis is considered the most critical cancer biomarker among the various hallmarks [13] (Figure 2). These postulates are supported by the unusual co-existence of viral particles and the suppression of programmed cell death in cellular and subcellular architecture [65]. These viral particles cause the synthesis of a diverse set of molecules that cause prolongation or inhibition of apoptosis [30]. For instance, the EBV produces BHRF1 and BALF1 peptides, which act on Bcl2 and other anti-apoptotic proteins to prevent programmed cell death. In a recent study [66], found that pro-apoptotic genes similar to the BIM gene are competitively inhibited by EBV mi-RNAs and peptides similar to the BHRF1 and BALF1 peptides (Figure 2).

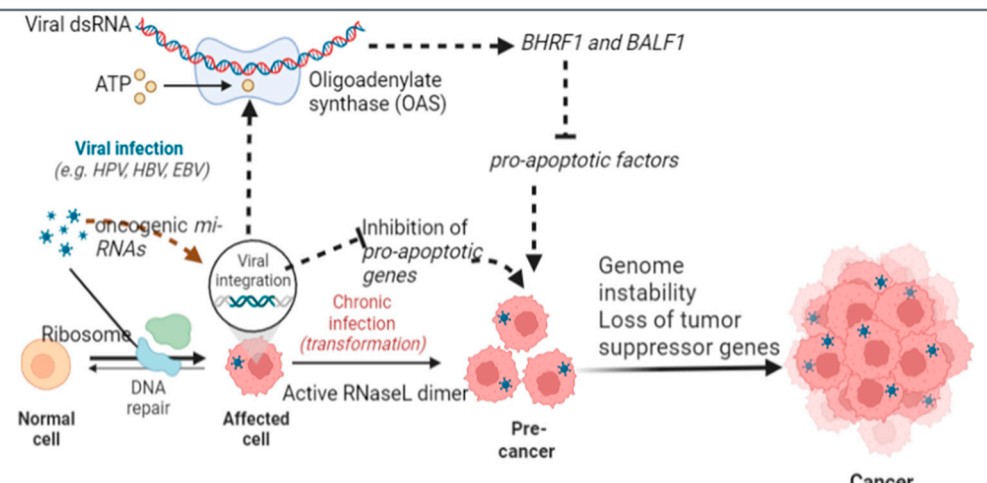

**Figure 2.** Mechanistic alteration in cellular and subcellular architecture of host cell caused by oncogenic viruses which inhibit programmed cell death and cause transition of a normal cell to cancerous cell (Image created by biorender.com, accessed on 2 November 2022).

Similarly, it has been found that other viruses, such as KSHV, disrupt pathways involved in apoptosis. For instance, viral constituents cause inhibition of ORF16 required for activation of apoptosis. Similarly, MCPyV causes overexpression of BIRC5/survivin mRNA production and inhibits caspase-mediated apoptosis [67]. Additionally, EBV leads to synthesizing of the HBV-HbX protein with a BH3 tail that irreversibly binds with Bcl2 and Bcl-xL to suppress apoptosis [68]. Although most classical pathways involve the role of Bcl2, recent studies have identified the upregulation of GRP78, suppression of pro-apoptotic genes (Bid and Bim), and dampening of endoplasmic reticulum stress pathways responsible for the promotion of apoptosis [89].

### 5.2. Reprogramming of Cellular Metabolic Pathways

Cancer cells undergo a paradigm shift in metabolism from aerobic to anaerobic respiration and utilize alternative metabolites for cellular metabolism, and this process is known as the Warburg effect [90]. For instance, viral proteins (E6 and E7) cause the upregulation of glucose transporter-1 and glucose transporter-4 [91]. Once glucose is internalized, E6 causes activation of mediators of the Warburg effect, while E7 potentiates the Warburg effect by irreversibly binding with pyruvate kinase M2 and thus accelerates the rate of glycolysis [38] (Figure 3). In concurrence with these findings, MCPyV has been found to cause increased glucose utilization by cancer cells and lactate production (an indirect biomarker of the Warburg effect). These metabolic transitions have been attributed to transcriptomic changes in various signaling pathways, such as mTORsignaling [32] (Figure 3).

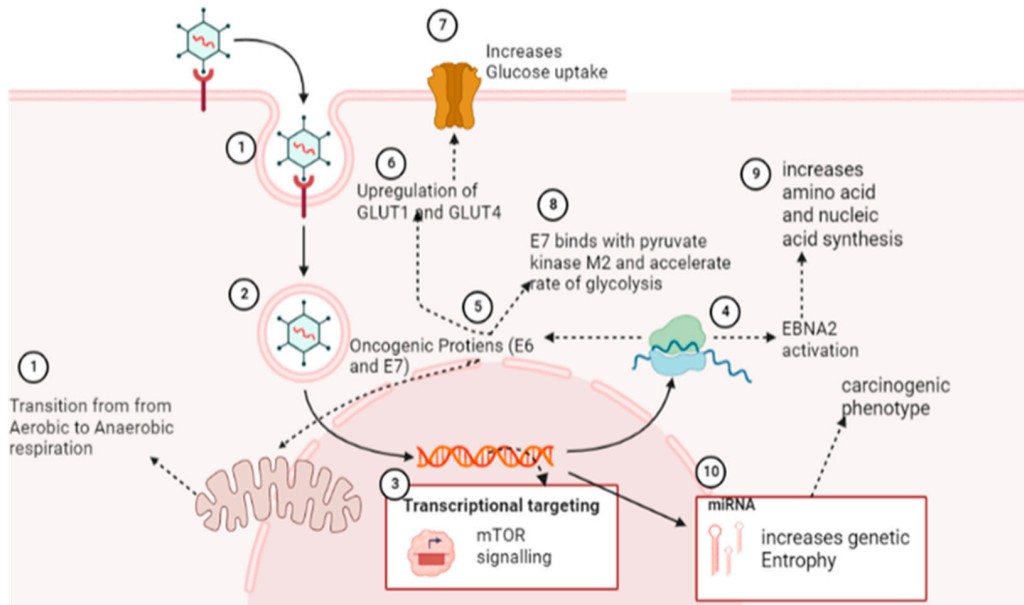

**Figure 3.** Transitions induced by oncogenic viruses in cellular metabolism make conditions favorable for virus and viral-induced cancer progression. This mechanism causes starvation of healthy cells under stressful conditions of the cancer microenvironment (Image created by biorender.com, accessed on 2 November 2022).

Through the activation of EBNA2, which triggers a myc-dependent metabolic program to increase amino acid and nucleotide synthesis, EBV infection promotes the adhesion and proliferation of B cells [72]. Viral infections also accelerate lipid metabolism, which is necessary for the de novo synthesis of new cells. In addition to increasing the production of proteins and nucleic acids, miRNAs of KSHV alter the metabolic pathways of cancerous cells from oxidative phosphorylation to glycolysis [31]. From these findings, it can be postulated that cancerous viruses cause alteration in cellular metabolic pathways by targeting multiple sites of metabolic pathways (Figure 3) [92].

In thermodynamics, entropy is the property of complex systems with multiple microstates, and the same applies to complex biological systems. Viral infection increases the number of microstates in biological systems and hence enhances genetic entropy [47]. This genetic entropy recons gene expression and other carcinogenic pathways by providing signal noise to enable host genomic determinants to configure into the lowest energy state or alternative energy state of the carcinogenic phenotype (Figure 3) [93]. Waddington landscape has provided a conceptual framework that explains the transition to an oncogenic state from the GRN state and has further proposed that research should be directed to understanding the thermodynamics of viral cancers and its utility in understanding carcinogenesis and cancer phenotype [48].

### 5.3. Transition of Cellular Microenvironment

The co-existence of viral-infected cancerous cells that survive under harsh environmental conditions is characterized by low oxygen levels and acidified environment [93]. Low oxygen levels around the cancerous microenvironment are attributed to the low vascularity and crowding of cancerous cells, which result in accelerated viral gene expression and, consequently, cellular stress response [26,94]. Hypoxic conditions result in the upregulation of hypoxia-inducible factors (HIF1A), which act on the genomic network to trigger the synthesis of VEGF, resulting in angiogenesis. It has been reported [95] that hypoxia causes localized immunosuppression, and oncogenic viruses attain the advantage of this phenomenon by inducing pseudo-hypoxia and hence avert immune cells from causing viral clearance [18]. Induction of pseudo-hypoxia and concurrently increased glucose utilization by cancerous cells starve immune cells and result in immune suppression [95]. Furthermore, hypoxia causes upregulation of EBV BRLF1, which decreases transcription of IRF3 and IRF7 genes responsible for interferon (INF-γ and TNFβ) production and down-regulates TLR9 mRNA [96]. These changes result in the alteration of the tumor microenvironment and hence the growth of cancerous cells and the proliferation of viral particles [97]. In this direction, various novel techniques and protocols have been used to understand the changes in the cellular microenvironment, and organoids offer an effective technique for an in-depth understanding of these changes [98].

Oncogenic viruses cause the synthesis of extracellular vesicles, which contain viral cargos, and these vesicles fuse with neighboring cells and result in the transmission of oncogenic viruses to neighboring cells [34]. Subsequent research found that before transmission of viral particles to neighboring cells, various metabolites such as miRNA and oncoproteins leach from cancerous cells. Recruitment of regulatory T cells causes localized immunosuppression and microenvironmental conditions conducive to the proliferation of cancerous viruses [39]. Oncogenic viruses induce heterogeneous phenotypes in host cells by causing alteration in host genome expression owing to the more flexible architecture of the viral genome, which enables them to move from one chromosomal compartment to another [39]. Similarly, the viral genome may be overexpressed or repressed with greater flexibility, resulting in switching from one cell type to another, known as cellular plasticity [99]. These characteristics of malignant cells enable them to develop resistance to immunotherapy, adapt to a broader range of environmental conditions, colonize new niches, spread infection, and avoid immune responses directed against tumor cells [66]. Furthermore, the highly unstable viral genome and its ability to mutate during the course of infection increase its propensity to cause cancer. From these findings, it can be postulated that oncogenic viruses cause alterations in the cancerous microenvironment, which causes the proliferation of cancerous cells at the expense of normal uninfected cells.

### 5.4. Modulation of Host Immune Response

Evolutionarily, the survival of predatory viruses and cancerous cells is only possible in an immune-compromised environment. Viral cancers are found to increase in an immune-compromised state induced by alterations and modulations of immune response by virus and their metabolites [100]. Oncogenic viruses cause upregulation of hotspots involved in immunosuppressive pathways by causing over-expression of PD-L1 and PD-L2, as well as the CTLA-4 [101] (Figure 4). In addition to this, a diverse set of viral components have been found to act on immunosuppressive pathways. These include: (i) downregulation of miR-34a, a prerequisite for the synthesis of PD-L1 [102] (ii) activation of NF-kB pathway by PD-L1 [103] (iii) inhibition of HLA expression by viral oncogenic proteins, which results in diminished recognition of tumor cells by the immune system and ineffectiveness of immunotherapy [104] (iv) Similarly, HBVsAg causes cell death in progenitor cells responsible for the production of various types of immune cells which further causes conditions favorable for carcinogenesis [35]. Similarly, viral oncogenic core proteins bind with host cellular gC1qRthatinhibits recruitment of immune cells towards cancerous cells and function as an efficient disabling mechanism of immune cells to cause viral and cancerous clearance [105].

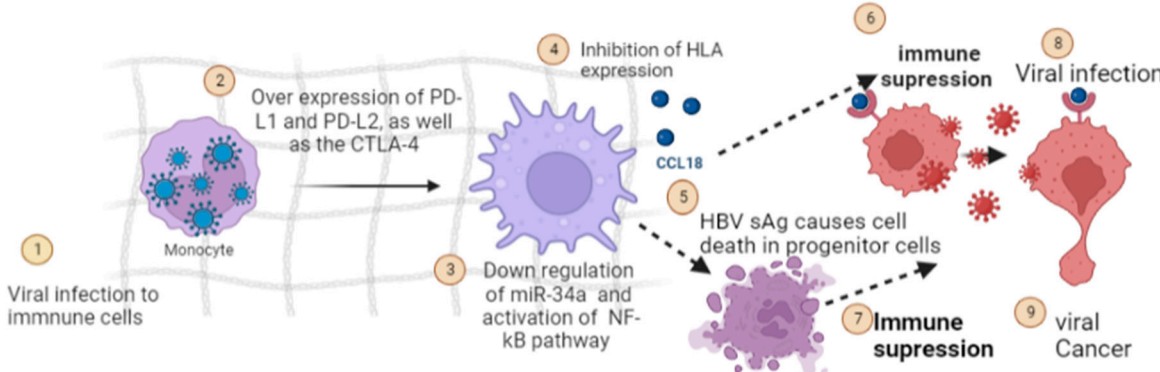

**Figure 4.** Immunosuppression is caused by oncogenic proteins of viral origin, which cause conditions favorable for the progression of cancer pathways induced by carcinogenic viruses (Image created by biorender.com, accessed on 2 November 2022).

### *5.5. Transcriptional Reprogramming*

Reprogramming of the transcriptional network is the hallmark of cancerousproliferation. It has been reported that early viral synthesized proteins cause dysregulation of the transcriptional network in favor of viral growth inside host cells. For instance, viral oncoproteins (E6, E7) and T-antigens are known to cause profound alteration in the transcriptional network of p53 and Rb-family complexes [106]. Viral oncoprotein (EBNA2) results in distortion in the tertiary structure of B-cell regulatory factors (RBPJ, EBF1, RUNX1, and PU1), which leads to the formation of cooperative DNA binding and creation of super-enhancers for cancerous growth. Additionally, EBNA2 results in the auto-activation of its transcription and other oncoproteins, further deteriorating B cells' functioning and cellular architecture. So all these alterations in the transcriptional network culminate in the down-regulation of tumor suppression genes (BIM and p16) [107]. The nuclear protein LANA, which is produced by the KSHV virus, is a prime example of transcriptional reprogramming because it leads to (i) structural and functional changes in the core histones H2A and H2B (ii) Binds to the GC-rich region of the chromosome via its C terminal (iii) Acts on the promoter region of the interferon regulatory factor and induces its overexpression, which reduces interferon synthesis. Although various critical points in pathways/hotspots of transcriptional reprogramming have been identified to be affected by carcinogenic viruses, the complexity of transcriptional pathways and diverse mechanisms involved in viral carcinogenesis is quite challenging for researchers to identify single and common casual factors [46]. For instance, studies have found that EBV-infected B cells result in hypermethylation of genes responsible for viral proliferation and causes directed expression of carcinogenic genes. These viral infections also cause suppression of cancer suppression genes [108,109] (Figure 5). Recently it has been found that HK2 causes the dose-dependent transition of human breast epithelial cells into mesenchyme cell type and enhances cellular migration into new niche areas. Based on this assumption, quantification of HK2-specific antibody levels in serum is proposed to serve as early biomarkers of breast cancer (Figure 1) [22].

### *5.6. Epigenetic Reprogramming*

Oncogenic Viral infection can cause epigenetic reprogramming through a diverse set of mechanisms by acting on various hotspots. For instance, the following epigenetic reprogramming pathways have been identified in viral cancers, including (i) Hyper- methylation of various genes responsible for establishing viral infection (HPV and EBV carcinomas). (ii) Hypermethylation results in the down-regulation of TET1 [27] and TET2, which are actively involved in demethylation at the deleterious site on the host genome (Figure 6). (iii) The activation of DNMT1 by viral oncoproteins (LMP1 and LMP2) results in hypermethylation at the CDH1 (E-cadherin) promoter region, which in turn results in the

downregulation of tumor suppressor genes (p16 and p21) [110]. (iv) Various viral oncoproteins cause structural and functional modification in histone proteins and hence directed expression of genes, especially Sting [111] and Apobec [112] pathways (v). Oncogenic viruses cause conformational changes in chromosomal architecture that leads to reorganization of DNA loops to ensure effective interaction of viral oncogenic proteins with the genomic framework. (vi) Oncogenic viruses cause chromosomal tethering, which decreases the proximity of the promoter region with the exon region and causes the transition of heterochromatic DNA into euchromatic DNA, which results in the opening of genomic territories and hence causes activation of cancer-related genes [113].

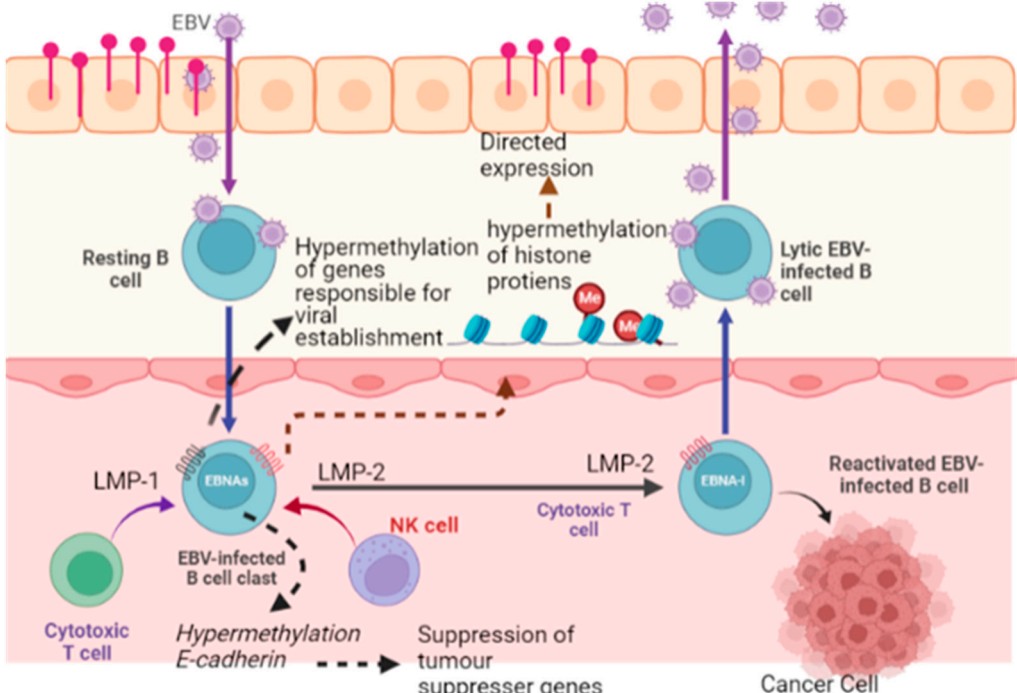

**Figure 5.** Transcriptional modification induced by carcinogenic viruses in the host genome causes activation of various carcinogenic genes and suppression of tumor suppression genes (Image created by biorender.com, accessed on 2 November 2022).

Oncogenic DNA viruses such as EBV have viral-specific programs designed to ensure the persistence of viral particles inside proliferative cells, which act as clonal outgrowths for cancer cells. These viruses encode proteins that protect viral DNA from degradation caused by cellular enzymes and non-enzymatic processes. Integration of viral genome with host cell genome occurs at specific sites of the host genome. For example, HBV viral genome has been observed to integrate at oncogenic hotspots such as TERT (telomerase), ERBB2, and PTPN13 loci [114]. Based on these observations, it can be postulated that infection with an oncogenic virus results in the transformation of the host epigenome or regulome. One of the significant limitations of these studies is that most of these findings are reported from preclinical studies and are very difficult to demonstrate in naturally occurring cases [115].

### 5.7. Host Factors for Viral Oncogenesis

Viral cancers are mostly descendant cells derived from progenitor cells infected with oncogenic viruses [116]. However, the researcher has found that for the successful establishment of viral infection and subsequent carcinogenic potential, the necessary prerequisite is to have a favorable mutation in host cells [117]. For instance, mutations in cell cycle control genes such as RASSF1A and CDKN2A1 are required to transform normal cells into EBV-associated NPC somatic cells. This result in the loss of function of the cell cycle control proteins 9p21 and 3p21.3, which then stabilizes telomerase and causes uncontrolled

proliferation of host cells. Another example of host variability is PI3K/MAPK mutation, which creates ideal conditions for hypermethylation at the promoter region of the NF-kB pathways and induces cellular damage that leads to cancer development [38]. Similarly, a positive correlation has been found between the incidence of viral cancers and mutational copies at HLA loci, suggesting that the processing of viral antigens plays a key role in viral cancers. Other mutations that enhance the susceptibility of host cells to viral carcinogenesis are presented in Table 1.

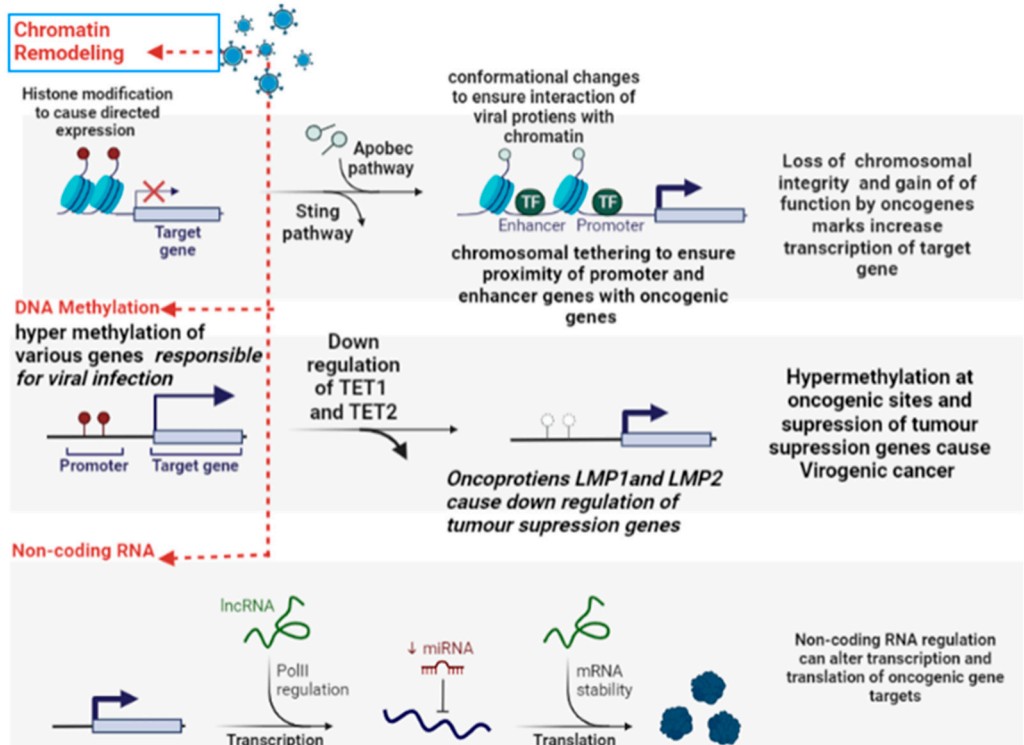

**Figure 6.** Epigenetic modification induced by oncogenic viruses which cause methylation at promoter region and enhancer region of oncogenes, chromatin remodeling, and other epigenetic changes favorable for Cancer (Image created by biorender.com, accessed on 2 November 2022).

In addition to host factors, environmental factors also contribute to viral carcinogenesis. For instance, co-infection of a host cell with the malaria-causing organism and HIV results in alterations of an immune response directed against virally infected cancer cells [118] (Figure 7). As per population genetics, cellular heterogeneity ensures viral survival in dynamic and stressed environmental conditions. Researchers have used energy landscape and canalization patterns to study cellular heterogeneity and viral cancer proliferation. They found that oncogenic viruses cause alterations in stable gene regulatory networks (GRN), resulting in greater cellular plasticity and hence heterogeneous cellular micro environment [109]. A recent study has found that the transition from a stable/uniform cellular state to a heterogeneous cellular state is promoted by increased signal noise in the GRN region. The viral genome has intrinsically higher signal noise than the host counterpart's genome, destabilizing GRN. Thus temporal variability in host microenvironmental factors, increased cellular heterogeneity, genetic variability, and immune functionality can significantly affect the progression of viral-induced carcinogenesis [119].

The late importance of the virome and microbiome on the skin and the mucous membrane has been identified as a risk factor for the progression of viral oncogenesis. Recent studies have reported that bacteriophage communities cause gut dysbiosis in colorectal cancer and make conditions favorable for establishing opportunistic organisms [120] (Figure 7). Subsequent studies have reported that virome signatures can be an essential marker for colorectal cancer. Furthermore, the presence of Lactobacillus in vaginal mucosa

offers protection against HPV infections and, subsequently, protection against the progression of vaginal cancer caused by HPV [121]. Integration may be a precursor to the passage from LSIL to HSIL, making it likely that it may serve as a biomarker for the development of cancer. Several chromosomal regions have integration hotspots, 3q28, 17q21, 13q22.1, 8q24.21, and 4q13.3 [22].

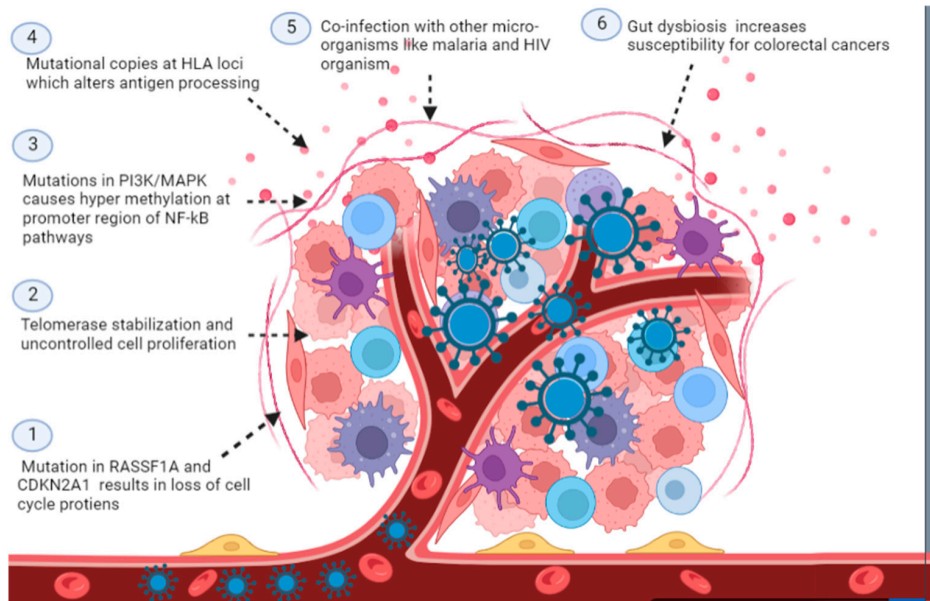

**Figure 7.** Host factors identified for the transition of normal cells into cancerous used by oncogenic viral products which have a role in the pathogenesis of viral cancers (Image created by biorender.com, accessed on 2 November 2022).

A commensal virome on the skin is beneficial against skin cancers through the mediation of the host immune system/response. It has been found that increased CD8 (+) T response against commensal viruses on the skin results in the development of vacant niche areas on the skin, which can be occupied by oncogenic viruses and subsequent cutaneous damage caused by chemicals and other physical factors results in the development of skin cancer [122].

### 5.8. Hijacking Anti-Viral Protective Mechanisms

Host cells contain an important regulator called Krüppel-associated box domain-containing zinc finger proteins (KRAB-KZFPs), a transcriptional product of this regulator that causes inhibition of retroviruses proliferation. The product inhibits retroviral multiplication by acting on the GC region of the retrovirus. However, under evolutionary pressure, these types of viruses have low GC content to evade a host protective mechanism that identifies significantly elevated levels of GC as foreign material and henceforth promotes its degradation. Furthermore, zinc-finger CCCH-type containing 11A (ZC3H11A) regulates RNA transport with the host cell. During infection with an oncogenic virus, this export mechanism is hijacked by viruses, and they use it for the transport of viral RNA cargo [22].

### 5.9. Role of Viral and Human miRNA in the Development of Cancer

miRNA is the non-coding RNA that plays a role in post-transcriptional modification of gene expression. Such miRNAs originating from a diverse set of oncogenic viruses result in a wide array of modifications in the host environment ranging from immune evasion to inhibition of apoptosis [123] (Figure 2). Among all viral miRNAs, EBV has been studied extensively. With the use of HTS technology, it has been observed that miR-BART19-5p is involved in B-cell tumors, miR-BART8-3p is involved in epithelial EBV-related tumors, and mir-BART16 in Burkitt lymphoma. Similarly, it has been observed that miR-31 causes

post-transcriptional regulation in TRIM8. It has been found to act as a critical player in Tumor Necrosis Factor-alpha and NF-kB signaling, which has implications in virus-induced gastric cancer. The interplay between viral circRNAs and host tumor suppression miRNAs has been established in various virogenic cancers [22]. EBV-encoded circRNAs are found to have a negative correlation with miR-203 and miR-31, which are tumor-suppressing genomic determinants. The mechanism by which viral miRNAs act on various pathways of host genomic determinants may be explained as the co-evolution of viral miRNAs with the host genome. The phenomenon is called red queen dynamics, meaning deep-time evolutionary events have resulted from the least energy thermodynamic balance between the host and viral genome. Common biochemical signatures identified in various oncogenic viruses include increased biosynthesis of host cellular miR-210, which regulates the post-transcriptional synthesis of Hypoxia Inducible Factor, which subsequently causes neo-angiogenesis and henceforth inhibits apoptosis [124].

*5.10. Human Endogenous Retroviruses (HERVs)*

HERV are the viral genomic elements introduced in the human genome almost 4.5 million years ago. These elements have exhibited co-evolution with the human genome and have transmitted across the generation following Mendelian inheritance. Most of these HERVs have been maintained under transcriptional control and have lost their function of replication and expression. The loss in functionality has been attributed to the long-term accumulation of mutations, frameshift mutations, and recombination culminating in the formation of solitary Long Terminal Repeats (LTRs) [125]. Due to evolutionary pressure, these LTRs have been imposed the physiological function of acting promoters and enhancers for host genes, and now they are considered junk DNA. These LTRs are considered important determinants of health and diseases due to their role as promoters and enhancers. In the following section, we review the role of HERV in various tumorigenic mechanisms [108].

Oncoviruses have been found to interact with HERV, which results in the expression of dormant HERV sequences [126] (Figure 8). These interactions cause epigenetic modification, chromatin disruption, and transcriptional alterations. A plethora of interactions have been reported between HERV-K HML2 (HK2) and oncoviruses; these interactions result in upregulation of HK2, which subsequently causes upregulation in rec and np9 and gag genes which are supposed to inhibit apoptosis and hence promotes carcinogenesis [127]. Furthermore, recent cell line studies have postulated that this interaction of endogenous and exogenous retroviruses causes immune evasion and promotes carcinogenesis.Furthermore, pathophysiology of Kaposi's sarcoma may be explained by the interaction of HK2 with the HIV genome, which causes immunosuppression through the activation of Np9 [128]. There are five viral proteins: trans activator of transcription, accessory protein negative factor Nef, matrix protein p17, envelope protein gp120, Reverse transcriptase RT and Tat. When secreted from HIV-1-infected cells, Gp120, Nef, p17, Tat, and RT produce oxidative stress and induce carcinogenic pathways [127,128]. Furthermore, these interactions are more prominent in older individuals than younger individuals, which explains the higher incidence of cancer in older individuals compared to its incidence in younger individuals [129].

HERV causes the reprogramming of somatic cells and induces Pluripotency characteristics in differentiated cells, and HERV-H is supposed to play a significant role in reprogramming. The role of HERV in carcinogenesis is because of their ability to cause the reprogramming of Cancer Stem Cells (CSC) [130]. HERVs are involved in various carcinogenic pathways, which render them essential candidates for carcinogenesis, and breast cancer is the most well-studied cancer in light of HERV.

HERV expression determines disease progression, the outcome of therapeutic intervention, and survival in different types of cancers. Colorectal cancer caused by activation of endogenous retroviral expression and concurrent CD8 (+) infiltration is very refractory to treatment. It has been observed that HERV-W causesthe cellular transition, cellular transformation, transcriptional alterations, and enhanced cell migration which increases cancer potential of Colorectal Cancer induced by endogenous oncoviruses [131]. Although, at the

present time, our knowledge of the role of HERVs in cancer progression is in the nascent stage, further research can provide novel therapeutic regimens against oncogenic cancers.

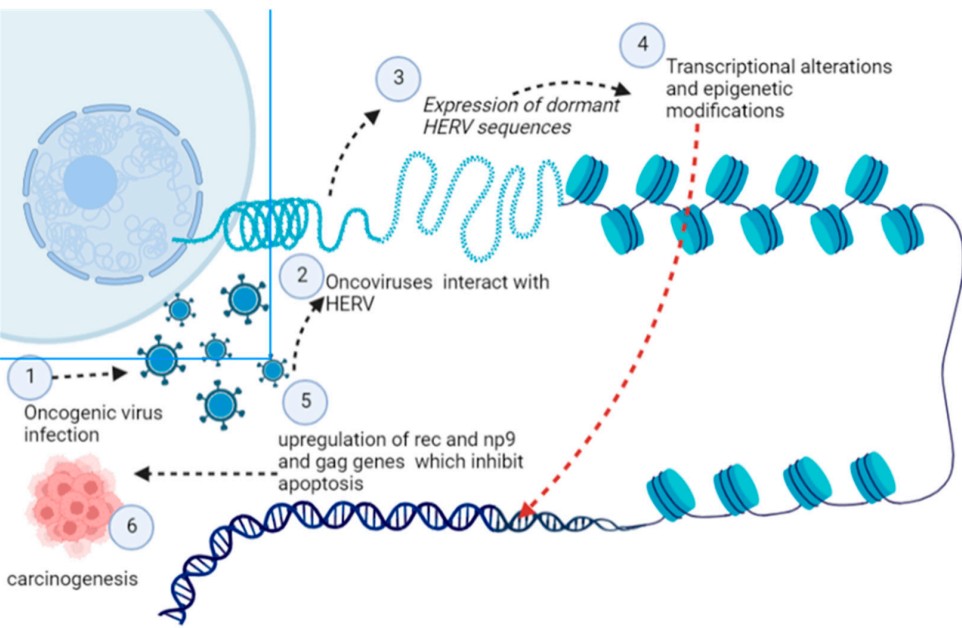

**Figure 8.** Activation of dormant HERV determinants induced by oncogenic viruses' which cause transcriptional, post-transcriptional, and epigenetic modification in the host cell for its transition into a cancerous cell (Image created by biorender.com, accessed on 2 November 2022).

### 6. Conclusions

Viruses are minute but complex organisms which cause a wide array of diseases in the animal kingdom, ranging from pandemics such as COVID-19 to various types of untreatable cancers. There is a lack of in-depth understanding of why some viruses are carcinogenic while others closely related are non-carcinogenic. From the findings of researchers across different fields of oncology, some common findings they highlighted include host factors such as immunosuppression, deleterious mutations, and Co-infection with other microorganisms. The common viral factors responsible for carcinogenesis include the ability of oncogenic viruses to attack various cellular pathways and perturb translational, transcriptional, and epigenetic pathways. These perturbations can have temporal and spatial heterogeneity and complexities, but still, they provide new insight into viral carcinogenesis and host determinants of the disease. Although various therapeutic alternatives are available against viral cancers, they lack efficacy, and their chronic use has been found to promote immune evasion by cancer cells. Hence, to develop an effective therapeutic regimen against viral cancers, a deeper understanding of mechanistic pathways involved in cancer carcinogenesis is needed. Understanding viral carcinogenesis can solve some critical questions about viral cancer biology, such as rate limiting steps involved, genetic hotspot, reversion of cellular microenvironmental conditions to retard the proliferation of cancer viruses, and the role of dampening genetic noise as a therapeutic intervention.

**Author Contributions:** Conceptualization, A.M.E.E., S.U.N. and O.S.S.; writing—original draft preparation, S.U.N.; writing—review and editing, S.M.B., U.M., S.I.A. and I.A.W.; visualization, N.A.N.A.; supervision, A.Y.E., A.A., F.O.A. and A.H.A.A.; funding acquisition, A.M.E.E. All authors have read and agreed to the published version of the manuscript.

**Funding:** The study was financially supported by the Department of Public Health, College of Health Sciences, Saudi Electronic University Public Health Saudi Arabia, Riyadh, 11673 Riyadh, Saudi Arabia.

**Conflicts of Interest:** The authors declare no conflict of interest.

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
