# Peer review of "Insight into Oncogenic Viral Pathways as Drivers of Viral Cancers: Implication for Effective Therapy"

_curroncol, doi:10.3390/curroncol30020150_

Round 1

Reviewer 1 Report (Previous Reviewer 2)

The authors made some improvements. However, some issues need to be addressed.

1. The punctuation marks. some commas are missing. Please carefully check the manuscript. For example, missing commas in line 35 after "however", lines 53, 60, 68, ... The English also need to be further improved.

2. line 586, should be section 6.

3. In table 1, the authors listed some oncogenic viruses with involved mechanisms (genes and signaling pathways). While in section 5, the authors discussed “Multiple mechanisms of viral carcinogenesis” which are mostly classified according to biological processes. These two parts did not match well. Section 5 may represent the influence of oncogenic viruses on the host cells. The authors should consider how to clarify the relationship of the biological processes, signaling pathways, and involved viruses more clearly.

Author Response

Response to Reviewer

Comment 1: The punctuation marks. Some commas are missing. Please carefully check the manuscript. For example, missing commas in line 35 after "however", lines 53, 60, 68, The English also need to be further improved.

Response: English language has been improved and commas and other punctuations have been added to manuscript. 
 Comment 2: line 586, should be section 6.

Response: Corrected as suggested in revised manuscript.

Comment 3: In table 1, the authors listed some oncogenic viruses with involved mechanisms (genes and signaling pathways). While in section 5, the authors discussed “Multiple mechanisms of viral carcinogenesis” which are mostly classified according to biological processes. These two parts did not match well. Section 5 may represent the influence of oncogenic viruses on the host cells. The authors should consider how to clarify the relationship of the biological processes, signaling pathways, and involved viruses more clearly.

Response: Corrected in revised manuscript, we have attempted to match viral carcinogenesis with biological processes involved in viral oncogenesis. 

This manuscript is a resubmission of an earlier submission. The following is a list of the peer review reports and author responses from that submission.

Round 1

Reviewer 1 Report

In this review, the authors introduced the complicated interactions between viruses and host and various mechanisms of viral-related cancers.  A large body of useful information was included and discussed. The structure of the text is logically clear and readable.

Main points:

1.     The authors proposed the possibility of oncogenesis activity of SARS-Cov-2 basing on the effects of SARS-Cov-2 on the expression of oncogenic genes and the induction of cellular activities like autophagy. This concept is interesting given numerous people have been infected by the virus in recent three years. But there is neither direct evidence nor detailed information about viral oncoprotein(s). It should be necessary to ask whether other coronaviruses such as SARS and MERS are associated with cancer in a similar manner.

2.     Do the DNA viruses and RNA viruses cause cancers by similar mechanisms?

3.     In table 1, HPV related cancer, HPV6 and HPV11 are listed as cancer causative viruses. In most cases, HPV6 and HPV11 are described as low-risk HPV and are mainly correlated with genital warts. In figure 1, HPV E6 and E7 are illustrated as oncogenic proteins in breast cancer and colorectal cancer. why these two types of cancer are not included as HPV-associated cancer in table 1?  

4.     In line 507, a citation is missed in the bracket.

Author Response

Reply to Reviewer Comments

Reviewer-1

Comment 1: The authors proposed the possibility of oncogenesis activity of SARS-Cov-2 basing on the effects of SARS-Cov-2 on the expression of oncogenic genes and the induction of cellular activities like autophagy. This concept is interesting given numerous people have been infected by the virus in recent three years. But there is neither direct evidence nor detailed information about viral oncoprotein(s). It should be necessary to ask whether other coronaviruses such as SARS and MERS are associated with cancer in a similar manner.

Response: Although there are no direct evidences but indirect evidences indicate role of SARS and MERS in cancer and same is discussed in detail in revised manuscript. 

Comment 2: Do the DNA viruses and RNA viruses cause cancers by similar mechanisms?

Response: Detailed information is included in revised manuscript and is highlighted with yellow color.

Comment 3: In table 1, HPV related cancer; HPV6 and HPV11 are listed as cancer causative viruses. In most cases, HPV6 and HPV1 1 are described as low-risk HPV and are mainly correlated with genital warts. In figure 1, HPV E6 and E7 are illustrated as oncogenic proteins in breast cancer and colorectal cancer. Why these two types of cancer are not included as HPV -associated cancer in table 1?

Response: Information about these (HPV6 and HPV11) is included in revised table of the manuscript and same has been highlighted with yellow color.

Comment 4: In line 507, a citation is missed in the bracket.

Response: Citation in line 507 has been included

Reviewer 2 Report

I would like to thank for the opportunity to review this paper.

 This review paper includes enormous information related to viral carcinogenesis, such as the mechanisms of viral carcinogenesis, ten viral oncogenic pathways. However, this paper suffers from considerable disorganization leading to a mess for readers.

 Here are some major comments:

 1. The usage of punctuation marks. Too many wrong usage of punctuation marks can be found in this paper. For examples, line 39-40: “Classical pathways of viral oncogenesis have identified oncogenic mediators in oncogenic viruses but these mediators have been reported to act on diverse cellular and multiple omics pathways.” There should be a comma before “but”. Line 45, a comma should be after “In present”. Many commas are missing in the manuscript.

2. Extensive editing of English language and style required. Such as the title, the “and” after colon should be removed. Line 173, “Multitude” is a noun, should it be “Multiple”. Line 248-249, Although and but should chose only one of them, and comma is missing…

3. The title of this paper is “Insight into oncogenic viral pathways as drivers of viral cancers: and implication for effective therapy”. However, most of this paper just presents the results from the literatures, not as the authors claimed in the abstract: “integrate concepts from system biology and cancer microenvironment, evolutionary perspective and thermodynamics to understand role of viruses as drivers of cancer”. In addition, the discussion part of “implication for effective therapy” is too general and simple. More specific discussion of therapy related with the content of this paper should be included for a comprehensive presentation.

4. The Figures and table in this paper did not match well with the manuscript. Such as Figure 1 in the main text is only one sentence “Furthermore there are accumulating evidences that support role of viral oncogenes in pathogenesis of breast cancer and colorectal cancer (Figure 1)” without detail information in either Figure legends or main text. All the figures were only cited once in the manuscript, leading to some confusions for the readers. The Table 1 listed some group I carcinogenic viruses, however, most of the information in this table lacks further explanation.

5. The mechanisms or viral oncogenic pathways and corresponding carcinogenic viruses are not well illustrated, given that the mechanism involved in table 1 did not match well with section 4.

6. Line 534 should be section 5, not 9?

Author Response

Reply to Reviewer Comments

Reviewer -2

Comment 1: The usage of punctuation marks. Too much wrong usage of punctuation marks can be found in this paper. For examples, line 39-40: “Classical pathways of viral oncogenesis have identified oncogenic mediators in oncogenic viruses but these mediators have been reported to act on diverse cellular and multiple omics pathways.” There should be a comma before “but”. Line 45, a comma should be after “In present”. Many commas are missing in the manuscript.

Response: we have revised manuscript as suggested by reviewer and have used punctuations correctly in revised manuscript.

Comment 2: Extensive editing of English language and style required. Such as the title, the “and” after colon should be removed. Line 173, “Multitude” is a noun, should it be “Multiple”. Line 248-249, although and but should chose only one of them and comma is missing.

Response: we have attempted for editing of “English language and style” and have incorporated corrections suggested.

Comments 3: The title of this paper is “Insight into oncogenic viral pathways as drivers of viral cancers: and implication for effective therapy”. However, most of this paper just presents the results from the literatures, not as the authors claimed in the abstract: “integrate concepts from system biology and cancer microenvironment, evolutionary perspective and thermodynamics to understand role of viruses as drivers of cancer”. In addition, the discussion part of “implication for effective therapy” is too general and simple. More specific discussion of therapy related with the content of this paper should be included for a comprehensive presentation.

Response: For system biology and cancer microenvironments please refer to “section 5.3, 5.4 and 5.5”. For evolutionary perspective please refer to “section 5.10”. For thermodynamics please refer to “Section 5.2 and 5.9”. For implication for effective therapy we have added contents under different sections.

Comment 4: The Figures and table in this paper did not match well with the manuscript. Such as Figure 1 in the main text is only one sentence “Furthermore there are accumulating evidences that support role of viral oncogenes in pathogenesis of breast cancer and colorectal cancer (Figure 1)” without detail information in either Figure legends or main text. All the figures were only cited once in the manuscript, leading to some confusion for the readers. The Table 1 listed some group I carcinogenic viruses, however, most of the information in this table lacks further explanation.

Response Corrected in revised manuscript.

Comment 5: The mechanisms or viral oncogenic pathways and corresponding carcinogenic viruses are not well illustrated, given that the mechanism involved in table 1 did not match well with section 4.

Response: Corrected in revised manuscript

Comment 6:  Line 534 should be section 5, not 9?

Response: Line 534 has been moved to section 9 from section 5.
